# Plant-Programmed Cell Death-Associated Genes Participation in *Pinus sylvestris* L. Trunk Tissue Formation

**DOI:** 10.3390/plants11243438

**Published:** 2022-12-09

**Authors:** Yulia L. Moshchenskaya, Natalia A. Galibina, Kseniya M. Nikerova, Tatiana V. Tarelkina, Maksim A. Korzhenevsky, Irina N. Sofronova, Maria A. Ershova, Ludmila I. Semenova

**Affiliations:** Forest Research Institute, Karelian Research Centre of the Russian Academy of Sciences, 11 Pushkinskaya st., 185910 Petrozavodsk, Russia

**Keywords:** *Pinus sylvestris* L., programmed cell death, xylogenesis, heartwood formation, bifunctional endonuclease, cysteine endopeptidase, metacaspase

## Abstract

Molecular genetic markers of various PCD (programmed cell death) variants during xylo- and phloemogenesis have been identified for the first time in Scots pine under lingonberry pine forest conditions in Northwest Russia (middle taiga subzone). PCD is a genetically determined process. Gene profiles of serine and cysteine proteases (endopeptidases), endonucleases, and metacaspases families are often considered markers of the final xylogenesis stage. In the present study, we examined the gene expression profiles of the *BFN* (bifunctional endonuclease) family—*BFN*, *BFN1*, *BFN2*, *BFN3*, and peptidase (cysteine endopeptidase, *CEP* and metacaspase, *MC5*) in the radial row, in addition to the vascular phloem and cambium (F1), differentiating xylem (F2), sapwood (SW), and transition zone during the active cambial growth period of uneven-aged pine trees (25-, 63- and 164-cambial age (c.a.) years old). We have shown that the expression patterns of the PCD-related genes did not depend on the cambial age but were largely determined by plant tissue type. In the radial row F1-F2-SW, we studied the activities of enzymes, including sucrose in metabolism (sucrose synthase, three forms of invertase); antioxidant system (AOS) enzymes (superoxide dismutase, catalase); and peroxidase andpolyphenol oxidase, which belonged to AOS enzymes and were involved in the synthesis of phenolic components of cell walls. The activity of the enzymes indicated that the trunk tissues of pine trees had varying metabolic status. Molecular genetic PCD regulation mechanisms during xylem vascular and mechanical element formation and parenchyma cells’ PCD during the formation of Scots pine heartwood were discussed.

## 1. Introduction

Programmed cell death (PCD) is an actively regulated process of cell death. PCD mechanisms in plant organisms are observed in organ formation, cell differentiation, and plants’ response to pathogens [1,2,3,4,5]. Various PCD variants in plants are characterized by certain cytogenetic changes that are under the control of biochemical and molecular genetic markers [2]. The best known and best described PCD example in the plant development literature is the final stage of cambial cell differentiation [6,7,8]. The cambium is the lateral meristem that forms the conductive trunk tissues—xylem and phloem.

Phloem is the transport tissue of plants. The transport flow of assimilates through the phloem is maintained by differentiation of sieve elements, the main stages of which are as follows: strengthening of the cell wall, degradation of the nucleus and loss of most of the organelles, and formation of highly fluid cytosol. The residual organelles are localized directly under the plasma membrane. After nucleus degradation, the sieve elements of the phloem become dependent on neighboring cells that are closely associated with them (due to numerous plasmodesmata); these are named “companion cells” (in the gymnosperms—Strasburger cells) [9]. Thus, sieve elements of the phloem run differentiation, which is accompanied by partial autolysis of cellular content, without undergoing complete PCD [9,10]. It is believed that the differentiation of phloem cells begins under the influence of a transcription factor of the *MYB*(myeloblastosis) family ─ the *ALTERED PHLOEM DEVELOPMENT* (*APL*) gene [11]. The downstream *APL* targets are likely genes of the exonuclease family [12].

Xylem is water-conducting tissue that also has mechanical functions [13]. In coniferous plants, xylem consists of tracheal elements (TE) and parenchyma cells. Xylem formation includes mother cell formation, elongation, and secondary cell wall formation. The final step in the formation of xylem TE is PCD. The initial stages of xylem differentiation are regulated by NAC-domain transcription factors and VASCULAR-RELATED NAC-DOMAIN 1-7 (VND1-7). VND1-7 triggers the secondary cell wall deposition (over MYB transcription factors) and positively regulates proteases and nuclease (bifunctional endonuclease (*BFN*), metacaspase (*MC*), cysteine endopeptidase (*CEP*) gene family) [14]. The cell content of TE is degraded when hydrolytic enzymes are released into the cytosol from the broken central vacuole [15]. The key enzyme responsible for nuclear degradation in conductive elements of xylem is a bifunctional endonuclease that has both DNase and RNase activities [16]. The participation of endonucleases in the degradation of the nucleus during the formation of the phloem sieve elements has not previously been studied.

Along with tracheids, which undergo a complete PCD process, the xylem contains living parenchyma cells, which link the symplast and apoplast of woody tissues and perform important functions in various processes such as carbohydrate storage, water transport from roots to leaves, and wound response [17,18,19,20,21]. In *Pinus sylvestris*, a species that forms heartwood (HW), the cells of the radial parenchyma undergo significant cytological changes when moving toward the stem pith and, as a result, undergo complete PCD. Heartwood formation has several stages: dehydration of xylem (sapwood, SW) in the transition (from SW to HW) zone (TZ), formation of extractive substances, changes in cell wall structure, and destruction of the cell nucleus [21,22]. Molecular and genetic mechanisms of PCD in cells of the ray parenchyma are poorly understood. In previous work [23], a transcriptomic analysis of the trunk tissues of 46-year-old Scots pine trees showed high expression of the gene encoding a bifunctional endonuclease in TZ.

Xylem and phloem also differ in their composition of structural components and, as a result, are characterized by different metabolic statuses. The different activities of carbohydrate metabolism enzymes, antioxidant systems (AOS), and secondary metabolism enzymes characterize the tissues themselves and the processes occurring within them. Quantitative differences in the activity of these enzymes can serve as a marker of more intensive formation and/or consumption of certain substances of a carbohydrate and phenolic nature, a different balance of which is maintained both during PCD and during heartwood formation. Previous studies have shown that the rate of HW formation in *Pinus sylvestris* depends on the cambium age [24].

The aim of this work was to identify the molecular genetic markers involved in programmed cell death during the formation of various trunk tissues, using the example of *Pinus sylvestris* trees of different ages.

## 2. Results

### 2.1. Identification of BFN, CEP, and MC Genes in the Scots Pine Genome

#### 2.1.1. BFN Gene Family

A search of the Scotch pine genome revealed three genes encoding protein sequences homologous to BFN *Picea abies*. The *BFN* gene (TC158725) described in the article by Lim and colleagues [23], which was highly expressed in the transition zone during heartwood formation in *Pinus sylvestris*, was also used for further analysis. A structural analysis of the proteins showed that all sequences contained a domain specific to BFN (cd11010) and a signal peptide at the N-terminus. Comparative evolutionary analysis showed that all four Scots pine sequences clustered with S1-type endonucleases from other species (Figure 1).

#### 2.1.2. CEP Gene Family

A search of the Scotch pine genome revealed nine genes encoding protein sequences homologous to spruce papain-like cysteine proteases (PLCPs). A structural analysis of the proteins showed that all sequences contained domains that were specific to PLCPs (pfam00112, smart00848). Three sequences contained the smart00277 (granulin) domain.

Comparative evolutionary analysis showed that three sequences of Scots pine clustered with *CEP*, while the remaining six were homologous to other types of PLCPs (Figure 1). The *PSY00006946* gene was chosen for further analysis since its product was identified as the closest homologue of the *MA_34759g0010* product, the spruce cysteine endopeptidase gene, for which a high specific expression was shown in the PCD zone of spruce tracheid [25].

#### 2.1.3. MC Gene Family

A search of the Scotch pine genome revealed eight genes encoding protein sequences homologous to spruce MCs. A structural analysis of the proteins showed that all sequences contain domains that were specific to metacaspases (pfam00656).

Comparative evolutionary analysis showed that two sequences of Scots pine clustered with spruce and *Arabidopsis* type I metacaspases, while the remaining six sequences were homologous to type II metacaspases (Figure 1).

The *PSY00023144* gene product was identified as the closest homologue of *Picea abies* MC5. A high specific expression of the gene encoding MC5 was shown in the PCD zone of spruce tracheid [25]. Additionally, the product of the *PSY00023144* gene was the closest homologue of AtMC9, which has been shown to be involved in PCD during the differentiation of xylem vessels in *Arabidopsis*. In this regard, the *PSY00023144* gene was selected for further analysis.

### 2.2. Cytology

Several tissue samples were taken from each tree, including cells of trunk conducting tissues at different stages of development. Before performing biochemical and molecular genetic analyses, a microscopic analysis of samples of selected tissues was carried out.

Fraction 1 (F1), taken from the inner part of the bark, included cells of the cambial zone, conducting phloem, and a small portion of non-conducting phloem. Fraction 2 (F2), taken from the debarked surface of wood, included cells with differentiating xylem. This tissue layer contained expanding tracheid that had not yet lost their cellular content and cells of the ray parenchyma containing typical oblong-elliptical nuclei (Appendix A). In the inner layers of sapwood (SWin) and transition zone (TZ), the bulk of the tissue was composed of dead tracheid; living cells were represented by ray parenchyma and parenchyma sheath of axial resin canals. In the interior sapwood, nuclei in the cells of ray parenchyma were rounded off (Appendix A). In the transition zone, nuclei gradually disappeared; in some cells it was possible to observe unorganized chromatin left after the destruction of the nuclear membrane (Appendix A).

### 2.3. Metabolic Status

The activities of AOS and phenolic (SOD (superoxide dismutase), CAT (catalase), POD (peroxidase), PPO (polyphenol oxidase), PAL (phenylalanine ammonia-lyase), and carbohydrate metabolism enzymes (SS (sucrose synthase) ApInv (apoplastic invertase), CtInv (cytoplasmic invertase), VacInv (vacuolar invertase)) were determined in F1, F2, and SW of Scots pine trees at different cambial ages (c.a.) to determine the metabolic status of the studied tissues.

#### 2.3.1. Activity of AOS and Phenolic Metabolism Enzymes

In general, CAT activity did not change between F1, F2, and SW. The lowest SOD activity was observed in differentiating xylem (F2), while the lowest POD was observed in SW. A trend toward an increase in PAL activity was observed against the background of an increase in PPO activity in the radial row « F1-F2-SW» (Figure 2).

#### 2.3.2. Carbohydrate Metabolism

Enzymes of carbohydrate metabolism showed the following pattern: lower activity of SS and CitInv in SW compared with other zones was accompanied by higher activity of VacInv and ApInv (Figure 3).

#### 2.3.3. Principal Components Analysis Using Enzyme Activity

While conducting the investigation, we studied plants of different c.a.: 25-, 63-, and 164-year-old trees (five models of each age). Principal component analysis was carried out for a data set for enzyme activity values for trees of all studied age groups. Before the calculations, the initial data were standardized. Figure 4 shows that the studied plants were divided into three groups according to the studied tissues.

Principal components analysis (PCA) of data from 45 models reduced the 9 enzyme variables to 2 factors that captured 64.4% of the variation. Ordination of models by these factors showed three discrete distributions (Figure 4). F1, F2, and SW, which generally had low POD, CtInv, and SS and high PPO, VacInv, and ApInv, varied primarily in relation to factor 1 (45.2% of the variance, positively correlated with PPO, ApInv, VacInv, and PAL and negatively correlated with CtInv and POD). F1 and F2 varied along factor 2 (19.2% of the variance, positively correlated with SOD and CAT). Thus, the different metabolic status described the studied tissues. Of course, there were quantitative differences due to the age; however, in general, changes in the activity of the studied enzymes were similar in models of different ages. We can thus conclude that tissue specificity is the main feature.

### 2.4. Expression of PCD Genes in Scots Pine Trunk Tissue

#### 2.4.1. Expression of BFN, CEP, and MC Genes during Cambium Differentiation

Expression of the *BFN* gene family and *CEP* and *MC* genes was analyzed in F1 and F2 in uneven-aged (c.a. 25, 63, and 164 years) Scots pine trees. In F1, the highest level of expression was shown for the *BFN1* (significant differences compared to F2 were shown for trees of all ages) and *BFN2* (significant differences from F2 were shown for 25 c.a. years trees) genes among all the studied genes (Figure 5).

All the studied genes were expressed in F2. The highest expression in F2 was shown for the *CEP* and *MC5* genes. For 63 c.a. years trees, significantly, higher relative expression was shown of *CEP* and *MC5* in F2 compared to F1.

#### 2.4.2. Expression of Genes Related to Heartwood Formation

To identify molecular genetic markers involved in PCD during HW formation, we studied the expression of PCD genes (BFN gene family, CEP, MC5) in SWin and TZ.

Expression in SWin and TZ 25 and 164 c.a. years Scots pine trees was shown only for the BFN, BFN1, and BFN2 genes among all the studied PCD genes. The expression of these genes increased in TZ compared to SWin. Significant differences in expression levels between SWin and TZ were shown for the BFN (for 25 and 164 c.a. years) and BFN1 (for 25 c.a. years) genes.

The highest level of expression in the analyzed tissues was shown for the BFN gene. BFN gene expression in TZ was 12 (for 25 c.a. years) and 14 (for 164 c.a. years) times higher than expression in SWin (Figure 6).

## 3. Discussion

In this study, the expression profiles of the genes encoding BFN, CEP, and MC were studied for the first time during (1) the formation of phloem sieve elements; (2) PCD, which occurs during the differentiation of conductive and mechanical elements of the xylem; and (3) in the xylem ray parenchyma cells during HW formation in Scots pine.

Both the sieve elements of the phloem and xylem TE were differentiated from cambial cells; however, only the differentiation of TE included a program aimed at cell death [9]. TE development has been studied mainly in angiosperms, and very little is known about the course of these processes in conifers [26]. A model system in zinnia cell cultures (*Zinnia elegans* L.) was created [27] and it was possible to consistently describe the changes that occur with cells during TE formation using this system. Similar models were created for *Arabidopsis* (*Arabidopsis thaliana* L.) [27,28]. Synthesis and deposition of the secondary cell wall components lead to the cytoplasm density decrease, a change in the tonoplast permeability, which lead to vacuole increase and destruction [29]. Transcriptional regulators play a key role in triggering cambial stem cell differentiation and regulating secondary cell wall formation. Three types of regulators involved in regulating of gene expression of SCW biosynthetic genes (NAC domain master regulator (NO APICAL MERISTEM, ATAF1, ATAF2, and CUP-SHAPED COTYLEDON 2) and two MYB domain regulators) [30,31,32,33]. The NAC family (including VASCULAR-RELATED NAC-DOMAIN1 (VND1-VND7), NAC SECONDARY WALL THICKENING PROMOTING FACTOR 1 (NST1, NST2), and NST3/SECONDARY WALL-ASSOCIATED NAC DOMAIN PROTEIN 1 (SND1)) regulate SCW biosynthesis [34,35,36,37,38]. The transcription factor VND7 can regulate secondary cell wall formation by not only starting its formation program but also suppressing it through SUPPRESSOR OF ACAULIS 51(*SAC51*) [26]. A similar regulatory mechanism is also described for genes encoding cysteine peptidases (CEP), bifunctional endonucleases (BFN), and metacaspases (MC). Thus, VND7 activates the expression of genes encoding proteolytic enzymes to complete the process of differentiation of the TE of the xylem, named PCD (Figure 7). In *Arabidopsis*, cysteine peptidases XYLEM CYSTEINE PEPTIDASE 1 and 2 (XCP1 and XCP2) and metacaspase METACASPASE 9 (MC9) are important PCD participants in the formation of xylem TE [26]. They are involved in the autolysis of cellular content [39], while bifunctional endonuclease encoded by the BFN gene family is the key enzyme responsible for nuclear degradation in xylem vessels [16] (Figure 7).

We showed that all the studied PCD genes (*BFN* gene family, *CEP*, *MC5*) were expressed in the differentiating xylem in *Pinus sylvestris* plants of different ages. The highest expression was shown for the *CEP* and *MC5* genes (Figure 5). This may indicate the participation of these genes in autolytic processes during xylem TE formation in Scots pine trees. *CEP* and *MC5* genes are homologous to the *CEP* and *MC5* genes of *Picea abies*, whose high expression was shown in the PCD zone during the tracheid formation in spruce [25]. Significantly higher expression of *CEP* and *MC5* in F2 compared to F1 was shown in 63 c.a. year-old trees. 

Currently, there are little data on the molecular genetic control of the phloem cell differentiation program. In *Arabidopsis thaliana* plants, it was shown that the ALTERED PHLOEM DEVELOPMENT (*APL*) gene, which encodes a transcription factor of the MYB type, caused the acquisition of phloem identity. It has been shown that recessive *apl* mutation led to division disturbance and subsequent differentiation of phloem cells [11]. Further studies have shown that the downstream target of the *APL* gene can be the transcription factor NAC45/86 which, in turn, positively regulates the expression of genes of the *Arabidopsis* exonuclease family *(NEN 1-4)* [12] (Figure 7). The role of genes of the endonuclease family in the differentiation process of the phloem sieve elements has not been previously considered in the literature. In our study, we showed the expression of the *BFN* gene family and *CEP* and *MC5* genes in the cambial zone and differentiating phloem (F1) cells. Genes of the *BFN* family encode a bifunctional endonuclease with both DNase and Rnase activities and are involved in the degradation of the cell nucleus during PCD in plants [16]. Significantly higher expression values in F1 compared to F2 were shown for the *BFN1* (25, 63, and 164 c.a. years) and *BFN2* (25 c.a. years) genes (Figure 5), which likely indicates the participation of these genes in the nucleus destruction during the formation of phloem sieve elements.

More genes of the primary metabolism are up-regulated in bark/cambial regions compared to SW [40]. Decreases in CtInv activity in SW likely indicates a decrease in synthetic processes associated with a decrease in the proportion of primary metabolism in relation to secondary metabolism in SW. The distribution of oxygen and its active forms is of great importance when moving from the periphery to the trunk pith [41,42]. Thus, the highest SOD activity was observed in F1 tissues. The increased activity of SS in F2 was noticed probably due to the active use of sucrose in the differentiating xylem for the synthesis of the structural components of cell walls (cellulose). It is known that the SS form associated with the cytoplasmic membrane is part of the cellulose synthase complex, supplying UDP-glucose, which is formed during the sucrose breakdown for cellulose synthesis [43,44,45,46,47,48]. In addition, an increase in POD in the cells of the cambial zone may be associated with cell differentiation occurring in the xylem adjacent to the cambial layer and the lignification of the cell wall. POD isoenzymes are involved in both of these processes [49]. 

PCD, which occurs in the xylem ray parenchyma cells during HW formation is a process that has seldom been discussed in the literature. The cells of the xylem ray parenchyma undergo significant cytogenetic changes during HW formation. Regarding *Cryptomeria japonica*, Arakawa and colleagues [50] showed the changes that occur with the cells of the ray parenchyma as they move from the cambium to the stem pith. A large central vacuole containing a large amount of protein is replaced by small vacuoles with protein content and, after deformation, they lose vacuolar sap rich in peptides. The nucleus also undergoes changes while moving toward the HW, first changing its shape from fusiform to elliptical and then undergoing condensation (Figure 8). Dead parenchyma cells do not contain organelles. The molecular genetic mechanisms of PCD regulation during heartwood formation are also poorly understood. Transcriptomic analysis of 46-year-old Scots pine trees showed high expression of the *BFN* gene in TZ [23]. In our study, we showed an increase in the expression of the *BFN*, *BFN1*, and *BFN2* genes in TZ compared to the inner sapwood (SWin) layers. Significantly higher expression values in TZ compared to SWin were shown for *BFN* (25 and 164 c.a. years) and *BFN1* (25 c.a. years). It is likely that these genes in *Pinus sylvestris* are involved in nuclear degradation during HW formation.

As cells move toward the HW, significant metabolic rearrangements take place in the SW. SW cells are characterized by specific metabolic changes associated with the formation of tyloses and the biosynthesis of extractives such as phenolic compounds, lignin, and aromatic substances that accumulate in the vessels [50,51,52,53,54]. There is also an accumulation of reducing substances (mainly carbohydrates) in the SW [55] and part of the carbohydrates transported from the SW may be shunted into the biosynthetic pathway for polyphenolic compounds [56]. Therefore, it is likely that increases in PPO activity against the background of an increase in PAL activity—the main source of the formation of phenolic compounds—is associated with the synthesis of complex substances of a phenolic nature in SW, which subsequently form the chemical heartwood base [57,58]. Thus, for example, for *PAL*, transcript accumulation was highest in the exterior sapwood compared to differentiating xylem [59] and in the transition zone between SW and HW compared to the cambial zone [40]. Numerous peroxisomes found in SW can also indicate a high activity of PPO [60]. Increased activity of VacInv and ApInv in SW may lead to enhanced degradation of sucrose in the region where the synthesis and accumulation of phenolic heartwood extractives occur, as has previously been discussed in the literature [53,57,60].

## 4. Conclusions

We have considered molecular genetic aspects of the PCD process of trunk tissue formation of Scots pine trees. We identified the *BFN* gene family and *CEP* and *MC* genes in *Pinus sylvestris* genomes and studied patterns of PCD gene expression at phloem sieve elements formation (partial autolysis of cell content), xylem tracheid formation (complete PCD), and the PCD of ray parenchyma cells. We showed that the choice of the molecular genetic program did not depend on the cambial age but was determined largely by the type of plant tissue. We found some similarity between all studied processes (such as participation of *BFN* genes in the regulation of tissue development), but different gene isoforms had a tissue-specific expression pattern.

## 5. Materials and Methods

### 5.1. Study Objects

The study was conducted in the middle taiga subzone (Karelia Republic, Northwest Russia). We selected objects for the study in lingonberry pine forests, which are common in the northern and middle taiga. The pine age line was represented by three groups: (1) 30 years (cambial age 25 years)—the beginning of the vegetative-reproductive ontogenesis stage; (2) 70–80 years (the average cambial age 63 years)—the middle of the vegetative-reproductive ontogenesis stage; (3) 170–180 years (the average cambial age 164 years)—the age at the end of the vegetative-reproductive stage of ontogenesis and the beginning of the extinction stage of tree growth. On each site, five model trees were selected. The dominant trees without oppression signs and damage were selected as model trees.

### 5.2. Plant Sampling

The trunk tissue samples were taken during the cambial growth (21–22 June 2021) at breast height (1.5 m above ground level). For microscopic analysis blocks, including the phloem, cambial zone, and last two to three annual increments of wood, were cut out (5 × 5 × 3 mm, length × width × height). For molecular genetic analysis and determination of enzyme activity «windows», 6 * 8 cm were cut out of the trunk and the bark was separated from the wood. During cambial growth, the bark moves away from the wood along the expanding xylem zone. Tissue complexes (hereafter: Fraction 1) were prepared from the inner surface of the bark. The layers of tissue (hereafter: Fraction 2) were scraped off the exposed wood surface with a blade. The sampling of stem tissues was monitored under a light microscope. Then, cores were taken. The material was frozen in liquid nitrogen and stored at −80 °C. Under laboratory conditions, cores were illuminated in ultraviolet light for a few seconds to determine the heartwood boundary and samples were taken from the transition zone (TZ) (two growing rings at the border with heartwood) and internal sapwood (SWin) (two growing rings after TZ).

### 5.3. Microscopy

Stem tissue samples for microscopic analysis were fixed in 3% glutaraldehyde solution and stored at 4 °C until further processing. In the laboratory, wood samples were washed three times to remove glutaraldehyde with a phosphate buffer (pH 7.2–7.4) and passed through an alcohols’ series 30°, 50°, 70°. Then, the samples in 70 °C ethanol solution were placed in a refrigerator, where they were stored at a temperature of +4 °C until the preparation of the cross-section. Radial and transverse sections of 15–20 μm were made using a Frigomobil 1205 freezing microtome (R. Jung, Heidelberg, Germany). The sections were stained with 4% acetocarmine solution to identify nuclei [61]. Temporary slides were made with glycerol as a mounting medium. The number of biological replicates (sampled trees) for each age group was five. 

Microscopic analysis was carried out under an AxioImager A1 light microscope (Carl Zeiss, Jena, Germany) equipped with an ADF PRO03 camera. Images were processed with ADF Image Capture software (ADF Optics, Wuhan, China). The number of technical replicates was three for each sampled tree.

### 5.4. Gene Retrieval from the Scots Pine Genome by Bioinformatics Methods

The search for *BFN*, *MC* and *CEP* gene families was carried out using the *P. sylvestris* gene set in the GymnoPLAZA database (PLAZA Gymnosperms. Available online: https://bioinformatics.psb.ugent.be/plaza/versions/gymno-plaza/ (accessed on 8 June 2021)) For this purpose, known gene sequences (*BFN, CEP,* and *MC* for *Picea abies*) [25] and *Arabidopsis thaliana* were used.

Pine protein structure prediction was performed using National Center for Biotechnology Information (NCBI) (NCBI Conserved Domain Database. Available online: http://www.ncbi.nlm.nih.gov/Structure/cdd/cdd.shtml/ (accessed on 8 June 2021)) [62] and ScanProSite (ScanProsite. Available online: https://prosite.expasy.org/scanprosite/ (accessed on 8 June 2021)) [63]. 

Phylogenetic analysis was performed using the MEGA 7 program [64]. Multiple alignments for potential *BFN* of Scots pine and other species were performed using ClustalW. The phylogenetic tree was constructed using the Maximum Likelihood method based on the Kimura 2-parameter model with 1000 bootstrap replicates [65,66]. 

### 5.5. qRT-PCR

Isolation of total RNA was performed using an extraction CTAB buffer (pH 4.8–5.0); 100 mM Tris-HCl (pH 8.0), 25 mM EDTA, 2M NaCl, 2% CTAB, 2% PVP, and 2% mercaptethanol was added to the mixture before use. Separation of the aqueous and organic phases was conducted using a mixture of chloroform-isoamyl alcohol (24:1). RNA was precipitated using 25 mM LiCl; then, re-precipitation was carried out using an extraction SDS buffer: 1M NaCl, 0.5% SDS, 10 mMTris-HCl (pH 8.0), and 1 mM EDTA [67]. RNA was re-precipitated with absolute isopropanol.

Before the reverse transcription reaction, the resulting mixture was incubated with DNase (Syntol, Moscow, Russia) for an hour at 37 °C. The enzyme was inactivated during reverse transcription (RT) by heating the reaction mixture at 70 °C for 10 min. RT reaction was performed using «T100 Thermalcycler» (BioRad, Foster City, California, USA) with a set of MMLVRT reagents (Evrogen, Moscow, Russia) using Oligo(dT)15-and Random (dN)10-primer. The reaction mixture for PCR (25 µL totally) contained 5 µL qPCRmix-HS SYBR (Evrogen, Moscow, Russia), 1 µL of forward and reverse primers (0.4 µM) (Synthol, Moscow, Russia), 2 µL of template cDNA, and 16 µL of deionized, nuclease-free water. The final content of the cDNA reaction mixture for all samples was ~100 ng. qRT-PCR was performed under the following conditions: 95 °C for 5 min for a further 40 cycles, denaturation (95 °C, 15 s), annealing (52.7—61.6 °C, 30 s), and elongation (72 °C, 30 s). For each pair of primers, a negative control was used—PCR was performed in the absence of a cDNA template. The GAPDH gene was used as a reference gene for calculating the relative expression of genes, which, according to analysis using BestKeeper and NormFinder, was the only gene stably expressed in all tissues studied. The primer sequence used for qRT-PCR is shown in Table 1. 

The relative quantity of gene transcripts (RQ) was calculated from the formula:
RQ = E^−ΔCt^,(1) where ΔCt is the difference in the threshold cycle values for the reference and target genes, and E is the effectiveness of PCR. The effectiveness of PCR was determined individually for each reaction based on amplification fluorescence data using the LinRegPCR software (version 2021.1, dr. J.M. Ruijter, Amsterdam UMC, Amsterdam, Nitherlands) [68].

### 5.6. Enzyme Activity Analysis

Plant tissues were ground in liquid nitrogen to a uniform mass and homogenized at 4 °C in the buffer containing 50 mM HEPES (pH 7.5), 1 mM EDTA, 1 mM EGTA, 3 mM DTT, 5 mM MgCl_2_, and 0.5 mM PMSF. After a 15-min extraction, the homogenate was centrifuged at 12,000 *g* for 10 min (MPW-351R centrifuge, Poland). The supernatant was purified on 20 cm^3^ columns with Sephadex G-250. Aliquots with the highest protein amounts were collected. In tissues, the protein concentration was 10–50 μg/ml. Proteins in the extracts were quantified by a Bradford assay. The enzyme activity was determined spectrophotometrically using a SpectroStar Nano plate spectrophotometer (BMG Labtech, Ortenberg, Germany). The residue was assayed for CWInv and the supernatants for VacInv, CtInv, SS, SOD, CAT, POD, PPO, and PAL. The essay for invertases and SS was described in detail by Serkova et al. [69]. The assay for SOD, CAT, POD, PPO, and PAL was described in detail by Ershova et al. [70].

### 5.7. Statistical Data Processing

The results were statistically processed with PAST (version 4.0). Before starting the statistical analysis, raw data were initially tested for normality using the Shapiro-Wilk test. The significance of differences between variants was estimated by Mann–Whitney U-test. The significant difference was evaluated at the level of *p* < 0.05. Neighbor-joining clustering was used to select tree groups based on the expression of the studied genes.

All data in the diagrams appear as the mean ± SD, where SD is the standard deviation. Different letters indicate significant differences at *p*-value ˂ 0.05, according to the Mann–Whitney U-test results.

Principal component analysis was carried out for a data set for enzyme activity values for trees of all studied age groups. Before the calculations, the initial data were standardized.

The research was carried out using the equipment of the Core Facility of the Karelian Research Centre of the Russian Academy of Sciences. 

## Figures and Tables

**Figure 1 plants-11-03438-f001:**
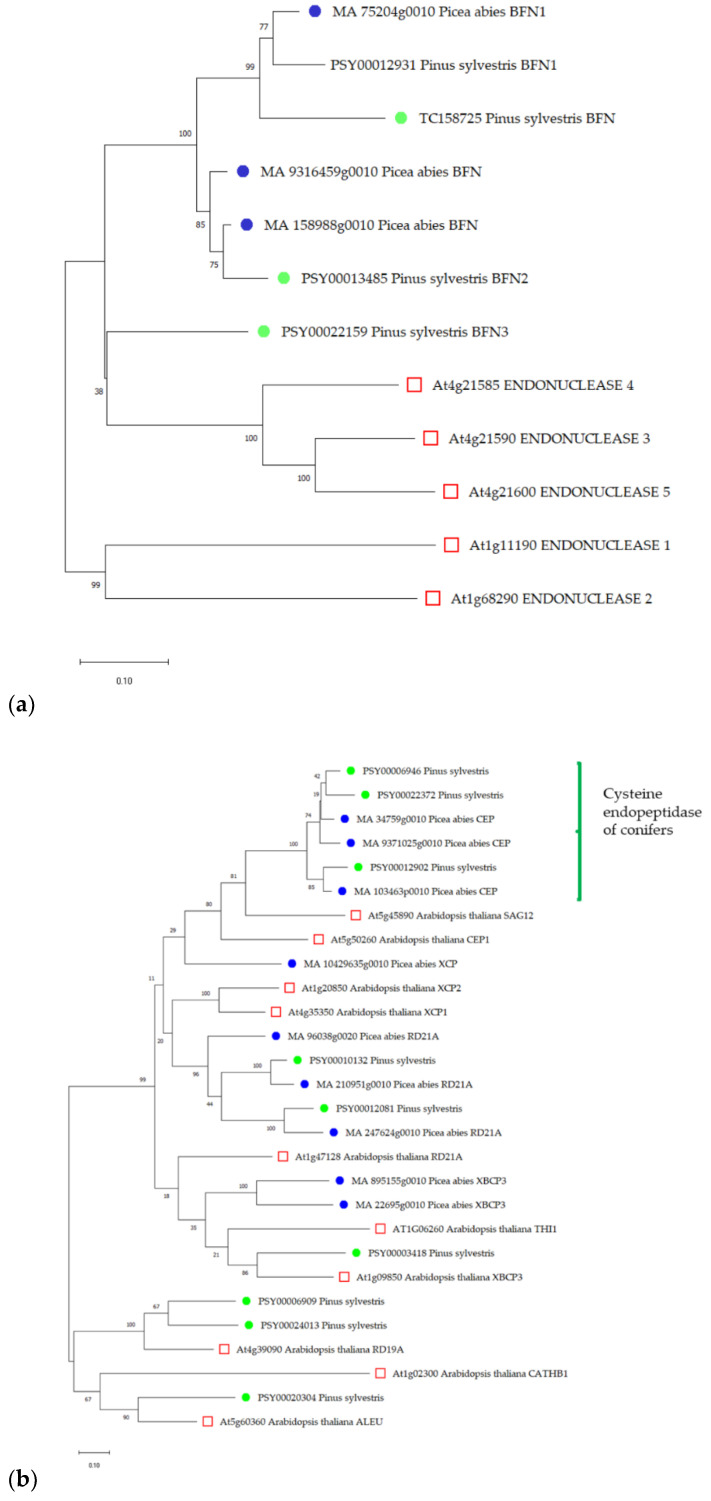
Maximum likelihood phylogenetic tree based on the nucleotide sequences of known genes from *Arabidopsis thaliana* L. (red squares), *Picea abies* L. (blue dots), and potential genes of *Pinus sylvestris* L. (green dots); (**a**)—bifunctional endonuclease (*BFN*); (**b**)—cysteine endopeptidase (*CEP)*; (**c**)—metacaspase (*MC*). The tree is drawn on a scale, with lengths of branches in the same units as the evolutionary distances used to infer the phylogenetic tree. The numbers shown next to the branches represent the results of the bootstrap test (1000 replicates). Along with the gene names are the database access codes: PLAZA (*P. sylvestris*), Congenie (*P. abies*), Phytozome (*A. thaliana*), and from Lim et al. [23] (*P. sylvestris BFN*, TC 158725).

**Figure 2 plants-11-03438-f002:**
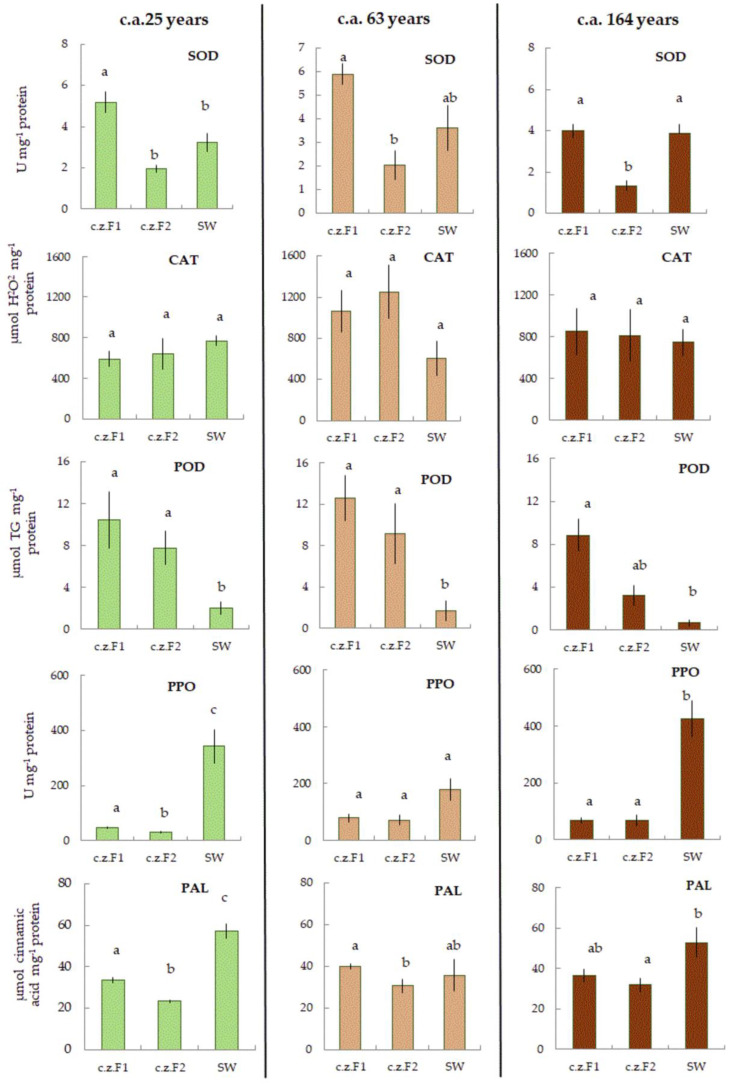
The activities of superoxide dismutase (SOD) (U mg^−1^ protein), catalase (CAT) (µmol H_2_O_2_ mg^−1^ protein), peroxidase (POD) (µmol TG mg^−1^ protein), polyphenol oxidase (PPO) (U mg^−1^ protein), and phenylalanine ammonia-lyase (PAL) (μmol cinnamic acid mg^−1^ protein) in Fraction 1 (cambial zone, conducting phloem, non-conducting phloem) (F1), Fraction 2 (differentiating xylem) (F2), and sapwood (SW) of 25, 63, and 164 cambial age (c.a.) years Scots pine trees. Values represent the mean ± SE (*n* = 5). The difference was considered significant if *p* < 0.05. Letters indicate differences between groups F1, F2, and SW.

**Figure 3 plants-11-03438-f003:**
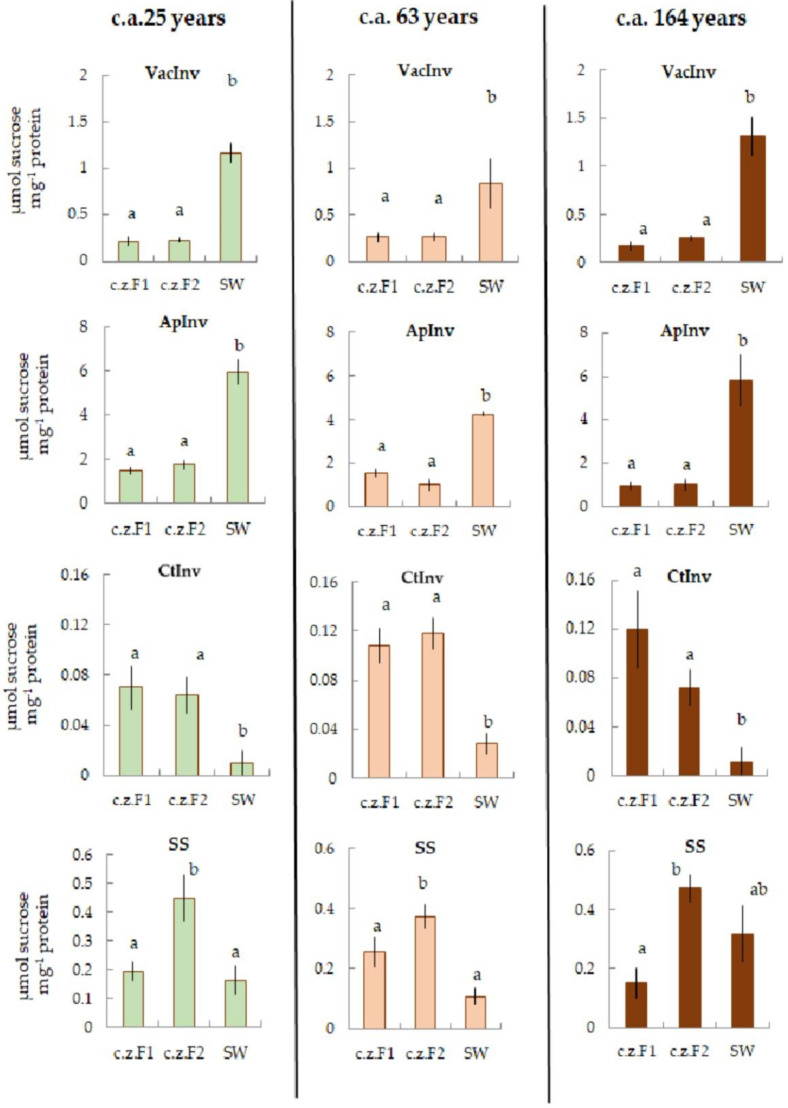
The activities of vacuolar invertase (VacInv) (µmol sucrose mg^−1^ protein), apoplastic invertase (ApInv) (µmol sucrose mg^−1^ protein), cytoplasmic invertase (CtInv) (µmol sucrose mg^−1^ protein), and sucrose synthase (SS) (µmol sucrose mg^−1^ protein) in F1, F2, and SW of 25, 63, and 164 c.a. years Scots pine trees. Values represent the mean ± SE (*n* = 5). The difference was considered significant if *p* < 0.05. Letters indicate differences between groups F1, F2, and SW.

**Figure 4 plants-11-03438-f004:**
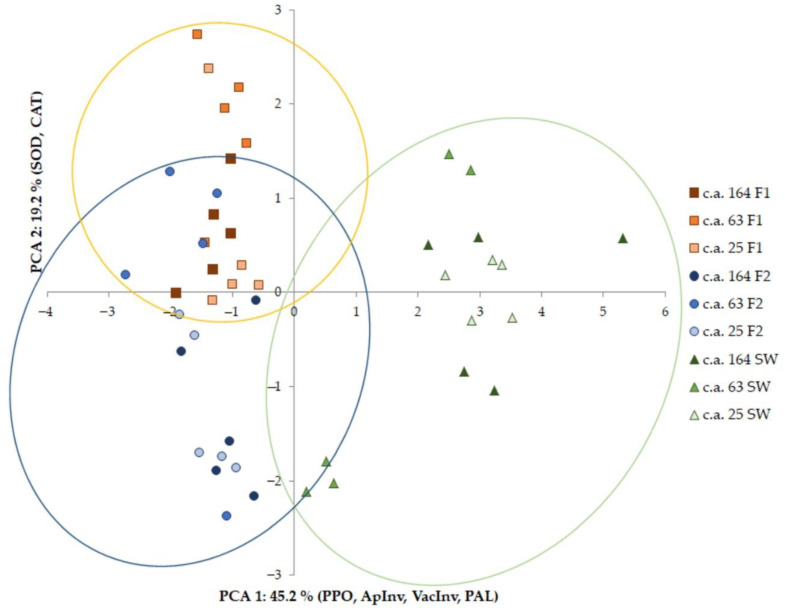
Ordination of 45 models based on enzyme activity. Factor 1 (45.2% of the variance) was correlated with PPO (r = 0.91), ApInv (0.92), VacInv (0.90), PAL (0.75), CtInv (−0.73), and POD (−0.62). Factor 2 (19.2% of the variance) was correlated with SOD (0.78) and CAT (0.66). The yellow grouping represents F1. The blue grouping represents F2. The green grouping represents SW. The values shown are means with 95% confidence intervals.

**Figure 5 plants-11-03438-f005:**
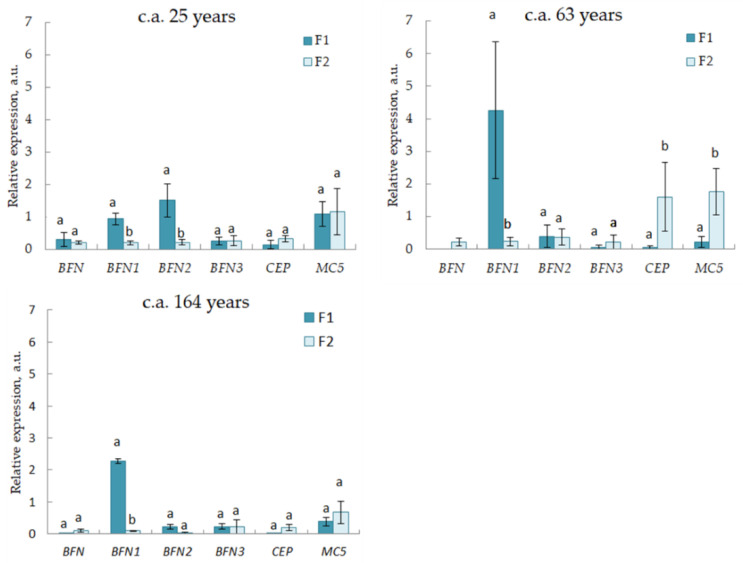
Relative expression (arbitrary units) of programmed cell death (PCD) genes in F1 and F2 of 25, 63, and 164 c.a. years Scots pine trees. Values represent the mean ± SE (*n* = 5). The difference was considered significant if *p* < 0.05. Letters indicate differences between F1 and F2 for each gene.

**Figure 6 plants-11-03438-f006:**
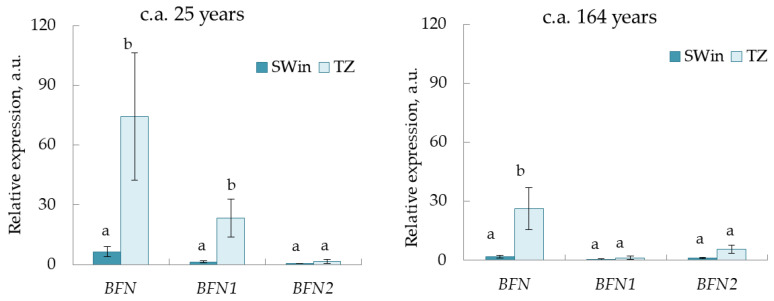
Relative expression (arbitrary units) of PCD genes in the transition zone (TZ) and inner sapwood (SWin) of 25 and 164 cambial age (c.a.) years Scots pine trees. Values represent the mean ± SE (*n* = 5). The difference was considered significant if *p* < 0.05. Letters indicate differences between SWin and TZ for each gene.

**Figure 7 plants-11-03438-f007:**
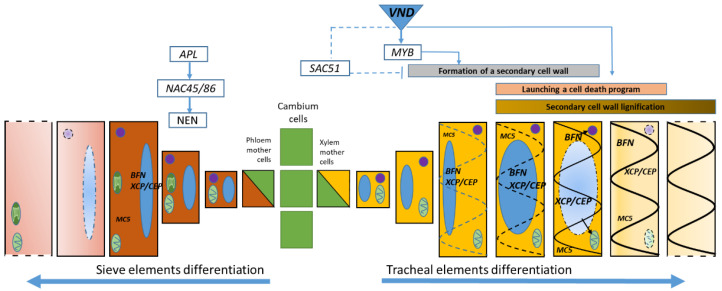
The cytology and molecular genetic control of differentiation of tracheal elements of the xylem (according to Escamez and Tuominen [26] with changes) and sieve elements of the phloem. The initial stages of xylem differentiation are regulated by the transcription factor VASCULAR-RELATED NAC-DOMAIN (VND), which, through the MYB (my elob lastosis) protein family, triggers the synthesis of secondary cell wall components or inhibits this process by triggering the SAC51 (SUPPRESSOR OF ACAULIS 51) suppressor. In addition, VND positively regulates the expression of genes encoding BFN, CEP/XCP, and MC, which are involved in the proteolysis of cell structures under PCD. Phloem cell differentiation begins under the action of the ALTERED PHLOEM DEVELOPMENT (*APL*) gene, whose downstream targets are NAC45/86 (NO APICAL MERISTEM, ATAF1, ATAF2, and CUP-SHAPED COTYLEDON 2) and nucleases of the NEN family [12]. Our study shows the possible involvement of genes of the *BFN* family in the process of differentiation of sieve elements of the phloem.

**Figure 8 plants-11-03438-f008:**
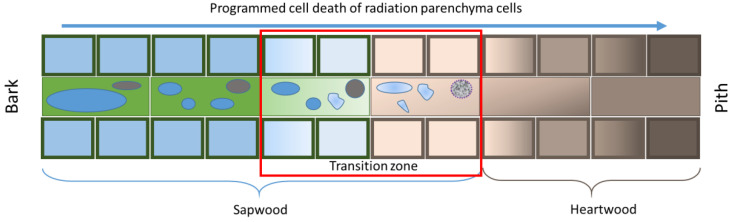
Scheme showing cytological changes in ray parenchyma cells as they move from the outer layers of SW to the PCD zone during HW formation. The outer layers of SW contain large central vacuoles and elliptical nuclei. As cells move toward the HW formation zone, the vacuoles are deformed and rounded nuclei then undergo degradation. The red box indicates the TZ.

**Table 1 plants-11-03438-t001:** Primer sequence for RT-PCR assays.

Gene Name	Citation/ID GymnoPlaza	Forward/Reverse Primer	Amplicon Length	Annealing Temperature, °C
*GAPDH*	PSY00009485	GGACAGTGGAAGCATCAT AACCGAATACAGCAACAGA	82	54.2 54.2
*BFN*	Lim et al., 2016	GGCTTACAAAGACGCTGAGG	158	53.8
		CTGAATCCCGAGTGTGGTCT		53.8
*BFN1*	PSY00012931	CCATAATGCCGAAGGAGAA	87	61.1
		GCTCTGCTGCCATAAGTT		61.6
*BFN2*	PSY00013485	AAGACGCTGATGAAGACA	96	60
		CCAACCTTACACCTCCTT		60
*BFN3*	PSY00022159	TGATGAGATTCGTTATTG	163	53
		CTGGTCAGTATAATTGTTA		52.7
*CEP*	PSY00006946	AAGGAATCAATTACTGGATAG TTCAACTGCTTCAATACC	95	56.5 56.3
*MC5*	PSY00023144	TAACGCTCTTCAATCAAT ATGCTGTGAGTATTCTTC	106	55.2 55.3

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
