# Peer review of "Plant-Programmed Cell Death-Associated Genes Participation in Pinus sylvestris L. Trunk Tissue Formation"

_plants, 2022, doi:10.3390/plants11243438_

Round 1

Reviewer 1 Report

The manuscript entitled "Plant programmed cell death associated genes participation in Pinus sylvestris L. trunk tissues formation" by Moshchenskaya et al. is a very interesting work concerning the analysis of gene and enzymatic activity related to programmed plant cell death in the formation of bark tissues in Pinus sylvestris, a tree species with a significant economic and ecological relevance. The work is well planned and executed, and relevant findings concerning the process under study are achieved. However, there are a few issues that prevent me from recommending its publication in its current form. First of all, English grammar and vocabulary should be thoroughly checked throughout the manuscript, as there are several expressions that are difficult to understand as well as several typos. Moreover, abbreviations should be explained the first time they are mentioned and used consistently thereafter.  I will comment the issues that I believe could be improved in the same order they appear in the manuscript:

- Abstract:  Though the abstract is OK, the authors do not state in it any of their major findings, which should be highlighted here.  Besides, authors should include the name of the species under analysis within the keywords.

 Minor mistakes: "..variants of during..."; it should be " ...genes expression profile..."; "..components of cell walls; as well". Please correct this last one.

- Introduction: this section is pretty much ok, however I missed a few comments concerning two key elements in the experiments, that is, the use of trees with different age and the analysis of enzymes related to sucrose metabolism. A brief paragraph should be added in which both these issues are introduced, and explain why they are relevant for the research.

Minor mistakes:"..tissue with water conduction..."; "...xylem is consisted of..."; "..radial parenchyma pass significant...", this last one it should say goes through.

- Results: as mentioned above, abbreviations should be explained the first time they are mentioned and used consistently thereafter. Please check and correct. In Fig. 1, subfamilies should be highlighted as they are later discussed in the text. In Line 134, I think Inner Sapwood is more correct than interior sapwood. The title of Section 2.3 is wrong, it should be "Metabolic status of tissues under study" or just "Metabolic status". At the beginning of this section, "...perform different functions that differ in composition...", certainly the tissues differ in composition, not the functions. Lines 146-149: this should go in Intro or Discussion. Lines 223-224: this is not what can be seen in the Figure, please correct.

 In the real time PCR experiments, it is not clear which sample is given value 1, or used as reference. Please clarify. Besides, I understand that arbitrary units refers to fold change, am I correct?

-Discussion: same issue with abbreviations. Here and in the Intro section authors discuss about transcription factors involved in the process under study, however no data concerning expression of transcription factors is analyzed in this work. I wonder if Figure 8 should be placed at the beginning of the Results section as it would help follow the results. Sometimes Discussion is a little bit reiterative, while at the same time  lacks comments discussing the results concerning sucrose metabolism and its putative implication in the process of PCD. Besides, statements in Lines 264-270 should be properly referenced.

Minor mistakes: "..required for phloem identification...", please correct, it should say "acquisition of phloem identity".  Mutants (apl) should be in italics. Line 331: "... a lot of protein...", please correct.  Line 332 :" The core also undergoes...", Which core?

Overall, this is a very interesting work. English grammar and vocabulary should be checked in all the manuscript. I encourage the authors to consider my comments and recommendations in order to see their valuable results published.

Author Response

Manuscript plants -2034642

«Plant programmed cell death associated genes participation in Pinus sylvestris L. trunk tissues formation»

Dear Ms. Averil Xu,

We are grateful to the reviewers for their analysis of our work and their comments.

We wrote the responses to the comments of the reviewers.

Sincerely,

Yulia L. Moshchenskaya, Natalia A. Galibina, Kseniya M. Nikerova, Tatiana V. Tarelkina, Maxim A. Korzhenevskii, Irina N. Sofronova, Maria A. Ershova, Ludmila I. Semenova

Reviewer 1 comments:

  1. …First of all, English grammar and vocabulary should be thoroughly checked throughout the manuscript, as there are several expressions that are difficult to understand as well as several typos.

Our response. We made the changes.

  1. …abbreviations should be explained the first time they are mentioned and used consistently thereafter.

Our response. We made the changes

  1. - Abstract: Though the abstract is OK, the authors do not state in it any of their major findings, which should be highlighted here.  Besides, authors should include the name of the species under analysis within the keywords.

Our response. We have added key findings for each section of the research and included the name of the species under analysis within the keywords.

  1.  Minor mistakes: "..variants ofduring..."; it should be " ...genes expression profile..."; "..components of cell walls; as well". Please correct this last one.

Our response. Thank you for your careful reading and recommendations. We tried to review the text more carefully. We've made changes.

  1. - Introduction: this section is pretty much ok, however I missed a few comments concerning two key elements in the experiments, that is, the use of trees with different age and the analysis of enzymes related to sucrose metabolism. A brief paragraph should be added in which both these issues are introduced, and explain why they are relevant for the research.

Our response. We've added information in «Introduction».

  1. Minor mistakes:"..tissue with water conduction..."; "...xylem is consisted of..."; "..radial parenchyma passsignificant...", this last one it should say goes through.

Our response. Thank you for your careful reading and recommendations. We tried to review the text more carefully. We've made changes.

  1. - Results: as mentioned above, abbreviations should be explained the first time they are mentioned and used consistently thereafter. Please check and correct. In Fig. 1, subfamilies should be highlighted as they are later discussed in the text. In Line 134, I think Inner Sapwoodis more correct than interior sapwood. The title of Section 2.3 is wrong, it should be "Metabolic status of tissues under study" or just "Metabolic status". At the beginning of this section, "...perform different functions that differ in composition...", certainly the tissues differ in composition, not the functions. Lines 146-149: this should go in Intro or Discussion. Lines 223-224: this is not what can be seen in the Figure, please correct.

Our response. - We have corrected the use of the abbreviations.

- We highlighted the CEP subfamily of conifers only, because no detailed analysis of other sequences belonging to other subfamilies was performed.

- We changed «interior sapwood» to «inner sapwood»

- We changed «Metabolic status of studying tissues» to « Metabolic status»

- We removed «…perform different functions…».

- We removed lines 146-149 from the «Result». Differences in enzyme activity in different variants of PCD are discussed in more detail in the "discussion" section.

-We removed the link to Fig. 6 from lines 223-224.

  1. In the real time PCR experiments, it is not clear which sample is given value 1, or used as reference. Please clarify. Besides, I understand that arbitrary units refers to fold change, am I correct?

We did not use any of the samples as a reference one. We calculated the relative expression based on the threshold cycle values for the reference and target genes. The formula for calculating relative expression is given in the «Materials and Methods».

  1. Discussion: same issue with abbreviations. Here and in the Intro section authors discuss about transcription factors involved in the process under study, however no data concerning expression of transcription factors is analyzed in this work. I wonder if Figure 8 should be placed at the beginning of the Results section as it would help follow the results. Sometimes Discussion is a little bit reiterative, while at the same time  lacks comments discussing the results concerning sucrose metabolism and its putative implication in the process of PCD. Besides, statements in Lines 264-270 should be properly referenced.

Our response. - We have corrected the use of the abbreviations.

- In «Introduction» and «Discussion», we described TF characteristics that were involved in the regulation of cambial differentiation. In our opinion, an important point is that the differentiation of cambial cells towards xylem and phloem is regulated by different transcription factors, but despite this, the downstream targets of these signaling pathways are the same PCD genes involved in the formation of both xylem and phloem. . In this regard, it seemed to us appropriate to include these TFs in the discussion.

- In the "Results" section, we don’t refer to Figure 8. At the beginning of the section, we give a small cytological characterization of the studied tissues with reference to Supplementary Figure 1. Therefore, we decided that it would be appropriate to place Fig. 8 in the "discussion" section.

- Since the article is mostly devoted to PCD genes, enzyme activity is given only as a characteristic of the metabolic status of tissues. In the "Discussion" section, the possible role of carbohydrate metabolism enzymes in PCD processes is considered. It is assumed that SS is included in the PCD during the xylem formation at the final stage (during the formation of the secondary cell wall), while ApInv and VacInv are involved in the sucrose degradation in the zone of synthesis and the accumulation of phenolic compounds during SW formation.

- We added links at Lines 264-270.

  1. Minor mistakes: "..required for phloem identification...", please correct, it should say "acquisition of phloem identity".  Mutants (apl) should be in italics. Line 331: "... a lot of protein...", please correct.  Line 332 :" The core also undergoes...", Which core?

Our response. Thank you for your careful reading and recommendations. We tried to review the text more carefully. We've made changes.

Reviewer 2 Report

1. Figure1 Re-layout to make it more beautiful.

2. I would like to suggest the red and green should not appear in the same picture.

3. In Figure 2, it is suggested to rearrange the layout to make the same indicators (such as SOD) of different ages more intuitive.

4. It is recommended to conduct similar rearrangement as Figure 2. 

Author Response

Manuscript plants -2034642

«Plant programmed cell death associated genes participation in Pinus sylvestris L. trunk tissues formation»

Dear Ms. Averil Xu,

We are grateful to the reviewers for their analysis of our work and their comments.

We wrote the responses to the comments of the reviewers.

Sincerely,

Yulia L. Moshchenskaya, Natalia A. Galibina, Kseniya M. Nikerova, Tatiana V. Tarelkina, Maxim A. Korzhenevskii, Irina N. Sofronova, Maria A. Ershova, Ludmila I. Semenova

Reviewer 2 comments:

  1. Figure1 Re-layout to make it more beautiful.

Our response. We changed Figure 2.

  1. I would like to suggest the red and green should not appear in the same picture.

Our response. We changed the colors in Figure 2 and Figure 4.?

  1. In Figure 2, it is suggested to rearrange the layout to make the same indicators (such as SOD) of different ages more intuitive.

Our response. We made the sign with the cambial ages in Figure 2.

  1. It is recommended to conduct similar rearrangement as Figure 2. 

Our response. We made the sign with the cambial ages in Figure 3 similar to Figure 2.
